# Laser Synthesis of Platinum Single-Atom Catalysts for Hydrogen Evolution Reaction

**DOI:** 10.3390/nano15010078

**Published:** 2025-01-06

**Authors:** Hengyi Guo, Lingtao Wang, Xuzhao Liu, Paul Mativenga, Zhu Liu, Andrew G. Thomas

**Affiliations:** 1Department of Materials, School of Natural Science, The University of Manchester, Oxford Road, Manchester M13 9PL, UK; hengyi.guo@postgrad.manchester.ac.uk (H.G.);; 2The Photon Science Institute, The University of Manchester, Oxford Road, Manchester M13 9PL, UK; 3Department of Mechanical and Aerospace Engineering, The University of Manchester, Oxford Road, Manchester M13 9PL, UK; 4Ningbo Institute of Materials Technology and Engineering, Chinese Academy of Sciences, 1219 West Zhongguan Road, Zhenhai District, Ningbo 315201, China; 5The Henry Royce Institute, The University of Manchester, Oxford Road, Manchester M13 9PL, UK

**Keywords:** single-atom catalyst, laser manufacturing, hydrogen evolution reaction (HER)

## Abstract

Platinum (Pt)-based heterogeneous catalysts show excellent performance for the electrocatalytic hydrogen evolution reaction (HER); however, the high cost and earth paucity of Pt means that efforts are being directed to reducing Pt usage, whilst maximizing catalytic efficiency. In this work, a two-step laser annealing process was employed to synthesize Pt single-atom catalysts (SACs) on a MOF-derived carbon substrate. The laser irradiation of a metal–organic framework (MOF) film (ZIF67@ZIF8 composite) by rapid scanning of a ns pulsed infrared (IR; 1064 nm) laser across the freeze-dried MOF resulted in a metal-loaded graphitized film. This was followed by loading this film with chloroplatinic acid (H_2_PtCl_6_), followed by further irradiation with an ultraviolet (UV; 355 nm) laser, resulting in pyrolysis of H_2_PtCl_6_ to form the SAC, along with a further reduction of the MOF to form a Pt-decorated laser-induced annealed MOF (Pt-LIA-ZIF8@ZIF67). The Pt-LIA-ZIF8@ZIF67 catalyst with a Pt loading of 0.86 wt. % exhibited exceptionally high activity for the HER in acidic conditions. The atomically dispersed Pt on the carbon substrate exhibited a small overpotential of 68.8 mV at 10 mA cm^−2^ for the hydrogen evolution reaction with a mass activity 20.52 times that of a commercial Pt/C catalyst at an overpotential of 50 mV vs. RHE. Finally, we note that the synthesis method is simple, fast, and versatile, and potentially scalable for the mass production of SACs for electrocatalytic applications.

## 1. Introduction

The increasing global demand for energy and resources has contributed to a substantial increase in the consumption of fossil fuels. This in turn has accelerated anthropogenic climate change and led to adverse environmental issues [1]. Consequently, the generation of hydrogen via water electrolysis presents a dual solution to these challenges. In this context, the development of efficient, durable, and economically viable catalysts to facilitate the practical and sustainable implementation of the electrocatalytic hydrogen evolution reaction (HER) is important [2,3]. Among the electrocatalysts available for the HER, platinum (Pt)-based catalysts exhibit exceptional efficacy in acidic media, attributable to the optimal binding energy of the Pt-H bond. Nevertheless, commercial Pt/C, which serves as the predominant catalyst for HER, shows some instability under harsh operational conditions, thus constraining its use for large-scale application [4,5]. In answer to this, platinum-based single-atom catalysts (SACs) have gathered considerable academic interest. These catalysts offer distinct advantages over bulk and nanoscale Pt, including maximized atomic utilization and unique coordination environments, thereby effectively reducing the requisite platinum loading and consequently lowering the associated cost [6,7].

Metal–organic frameworks (MOFs) have been utilized as precursors to fabricate a variety of carbon-based nanomaterials [8]. Employing conventional high-temperature carbonization techniques in an inert atmosphere allows MOFs to be transformed into transition-metal-based composites. These composites typically feature metal nanoparticles encapsulated within a porous carbon matrix, and they have demonstrated superior electrocatalytic capabilities in water-splitting applications [9]. Recent advances have revealed that MOF-derived carbon substrates can also serve as effective templates for single-atom catalysts (SACs) [10]. These MOF-derived carbons are characterized by their high surface area, intricate porous architecture, and an abundance of nitrogen dopants, which collectively facilitate the anchoring of highly mobile metal atoms [8].

Despite some progress, the carbonization of MOFs and the production of well-dispersed isolated single atoms on MOFs face a number of challenges to make such systems technologically viable [11]. Firstly, the annealing process employed for the reduction of MOFs is not a simple process and requires high temperatures. An extended duration may cause the aggregation of metal particles and result in the generation of harmful gasses. Additionally, the carbonization procedure is typically conducted within the spatially restricted confines of a tube furnace under an inert atmosphere. This leads to a fabrication process for electrode materials with constrained sizes, which is both time- and energy-intensive. Consequently, the scalable production of single-atom catalysts (SACs) remains limited due to factors such as procedural complexity, lack of versatility, prolonged processing durations, and elevated post-synthesis costs [12].

Laser-induced annealing has been used as a viable approach for the fabrication of a number of multifunctional materials, as it is relatively cheap and precise and allows ease of preparation, processibility, and scalability [13]. Although it is unlikely to be useful for large-scale heterogeneous catalysis manufacture, for electrocatalyst applications, laser processing is useful since the catalyst must be prepared on a substrate that allows it to be used as an electrode. Laser arrays or raster systems can therefore be used effectively to provide roll-to-roll processing of electrodes. The carbonization of materials using lasers has also been successfully deployed to synthesize carbon-based materials in ambient conditions [14]. This is possible because a laser can induce highly localized heating temperatures of over 2500 °C, which are capable of breaking C-O, C=O, or C-N bonds. A reaction of these elements yields gaseous by-products, which form a reducing atmosphere [15]. This reducing atmosphere, generated during the laser annealing process, then envelops the sample and facilitates the reduction of metal ions into metallic nanoparticles. At the same time, aromatic fragments in the organic framework reorganize to create graphitic carbon structures [16]. Utilizing a focused laser source, MOFs can be subjected to laser-induced treatments to yield derivatives possessing uniform structures. In these derivatives, metal nanoparticles can become encapsulated by a carbon shell, due to the speed of the carbonization process arising from the focused laser source under ambient conditions [17,18,19]. This laser-induced annealing strategy offers distinct advantages over traditional high-temperature annealing methods, including the prevention of particle aggregation, enhancement of yield, reduction in processing time, and potentially, reduced energy usage. Despite these merits, there exist only a few research articles exploring laser-annealed MOFs for water-splitting applications [19].

In this work, a two-step laser-induced annealing (LIA) strategy was employed to synthesize Pt single atoms on carbonized MOFs. Firstly, the MOF is irradiated with a focused 1064 nm wavelength infrared laser under ambient conditions to carbonize the MOF, which results in metal nanoparticles encapsulated in porous carbon. Irradiation with a UV marker laser is then used to synthesize Pt single atoms from a precursor molecule onto this carbonaceous material.

## 2. Materials and Methods

### 2.1. Materials

The chemicals used in this experiment were chloroplatinic acid hydrate (H_2_PtCl_6_·xH_2_O, 99.995%), Nafion (5% aliphatic alcohols and water), zinc nitrate hexahydrate (Zn (NO_3_)_2_·6H_2_O), cobalt nitrate hexahydrate (Co(NO_3_)_2_·6H_2_O), 2-methylimidazole (2-MeIm) (all Sigma-Aldrich, Merck, Darmstadt, Germany), and an ICP standard solution (SPEX CertiPrep, Metuchen, NJ, USA). All chemicals were used as received without further purification, except for dilution in deionized water (15 MΩ·cm).

### 2.2. Synthesis of ZIF-8

Briefly, 10 mmol of Zn (NO_3_)*_2_*·6H_2_O was dissolved into a mixture of 50 mL ethanol and 50 mL methanol to form solution A. Meanwhile, 50 mmol of 2-MeIm was dissolved into a mixture of 25 mL ethanol and 25 mL methanol to form solution B. Solution B was quickly poured into A with vigorous stirring, at 27 °C. After continuous stirring for another 24 h, the obtained precipitates were washed with a 1:1 *v*/*v* mixture of ethanol and methanol several times, and the precipitate was retrieved by centrifugation. The white powder of ZIF-8 was collected following drying overnight in an oven, at 60 °C.

### 2.3. Synthesis of ZIF-8/ZIF-67 Composite

The as-synthesized ZIF-8 (0.5 g) was transferred to 100 mL MeOH to form a suspension. Co (NO_3_)_2_.6H_2_O (5.82 g) and 2-MeIm (6.16 g) were each dissolved in 100 mL of methanol. The alcoholic solution of Co (NO_3_)_2_.6H_2_O was then added to the ZIF-8 suspension followed by the 2-MeIm solution. This resulted in the formation of a homogeneous dispersion. The solution was stirred for 24 h at room temperature, before the light-purple precipitate was collected by centrifugation, washed with MeOH, and dried by heating to 60 °C overnight.

### 2.4. Laser-Induced Annealing of MOFs on Glass Slides

In total, 12 mg of the MOF powder was dispersed in 0.4 mL of ethanol and ultrasonicated for 30 min. The as-formed MOF suspension was drop-cast onto one side of a glass slide of surface area 2 × 2.5 cm^2^ to obtain a ZIF-8@ZIF-67/substrate film with a mass loading of 3 mg cm^−2^.

An infrared laser system (central wavelength (*λ*_χentral_) = 1064 nm) (IPG Photonics, Coventry, UK), pulse width = 5 ns, repetition rate = 30 kHz, and average power = 10 W, was focused onto the ZIF-8@ZIF-67 film with a spot size ≈ 0.05 mm. The laser spot was rastered, with a line spacing of 0.5 mm, across the MOF film in a zigzag manner with a pattern dimension of 1 cm × 5 cm. The scanning speed of the galvo head was set to 300 mm/s, and the ZIF-8@ZIF-67 was subjected to laser beam patterning under Ar gas. The irradiation step was applied for 5 min and resulted in samples that will be referred to as LIA-ZIF8@ZIF67, hereafter.

The LIA- ZIF8@ZIF67 film was peeled from the glass slide. The collected film samples were then sonicated in water, washed three times using deionized water (15 MW cm), and dried by heating to 60 °C in a vacuum oven, overnight. Some of these films were kept for characterization, and some transferred to the next stage to produce the Pt-LIA-ZIF-8@ZIF-67 SACs.

### 2.5. Laser Synthesis of Pt Single-Atom Catalysts

The overall synthesis process of the Pt single-atom catalysts is illustrated in Figure 1. To prepare the Pt-LIA-ZIF-8@ZIF-67 film, 20 mL of 1 mg/mL LIA-ZIF-8@ZIF-67 dispersed in ethanol was vacuum-filtered through a PTFE filtration membrane (0.2 μm pore size, Merck Millipore, Darmstadt, Germany), resulting in an LIA-ZIF-8@ZIF-67 film of 3.2 cm in diameter and 1.5 ± 0.2 μm in thickness. In total, 1 mL of the metal precursor solution (H_2_PtCl_6_, 5 mmol L^−1^) was vacuum-filtered through the LIA-ZIF-8@ZIF-67-coated membrane and freeze-dried to form a chloroplatinic acid-infused LIA-ZIF-8@ZIF-67 film.

A UV laser system with *λ*_central_ = 355 nm, pulse width = 8 ns, repetition rate = 80 kHz, and average power = 3 W was focused with a spot size of ~0.02 mm, onto the chloroplatinic acid-infused LIA- ZIF8@ZIF67 film. The laser spot was raster-scanned across the film over an area of 3 cm × 3 cm with a line spacing of 0.02 mm to ensure the complete irradiation of the film. The galvo head scanning speed of the laser was set to 1000 mm/s. The dried chloroplatinic acid-infused LIA- ZIF8@ZIF67 membranes were subjected to direct laser beam irradiation under Ar. Finally, the resulting Pt-LIA-ZIF-8@ZIF-67 samples were peeled off from the filtration membrane, and the collected films were sonicated in DI water, and washed in deionized water (15 MW cm), three times. Finally, the films were freeze-dried to produce the catalyst powder used for characterization and catalytic activity measurements.

### 2.6. General Material Characterization

The core–shell ZIF-8@ZIF-67 before and after laser irradiation were studied by scanning electron microscopy (SEM) using a high-resolution scanning electron microscope (1 kV in lens mode, Zeiss Merlin, Zeiss, Oberkochen, Germany). High-resolution scanning transmission electron microscopy (HR-STEM) images of Pt-LIA-ZIF-8@ZIF-67 were captured by a C_s_-corrected microscope (FEI Titan G2 80–200 S/TEM ChemiSTEM (Thermo-Fisher, Walton, MA, USA)) operating at 200 keV. This instrument is also equipped with a high-efficiency Super-X EDS detector system allowing energy-dispersive X-ray spectroscopy (EDS) to be measured. To prepare TEM samples, powder samples were sonicated in ethanol to produce a dilute dispersion. They were then drop-cast onto a copper grid with a lacy carbon film. X-ray diffraction (XRD) was performed to determine the crystallinity of the catalyst samples (X’Pert Pro XRD5 diffractometer (Malvern PANalytical, Malvern, UK)). X-ray photoelectron spectroscopy (XPS) was carried out using a high-throughput XPS instrument (an ESCA2SR high-throughput X-ray photoelectron spectrometer) (Scienta-Omicron Uppsala, Sweden), fitted with a monochromatic Al Ka X-ray source (hn = 1486.6 eV), Argus CU multi-purpose hemispherical electron energy analyser, and low-energy electron flood gun to reduce the effects of surface charging, where necessary [20]. Spectra were calibrated on the binding energy scale relative to an adventitious hydrocarbon peak at 285 eV [21]. Curve fitting of XPS spectra was carried out using CasaXPS software [22]. Inductively coupled plasma–optical emission spectrometry (ICP-OES) (PlasmaQuant 9000 Elite, Analytikjena, Jena, Germany) was conducted to determine the metal content within the Pt-LIA-ZIF-8@ZIF-67. All samples were weighed and digested in aqua regia for two days, followed by dilution and filtration through a 0.2 μm pore size Whatman syringe filter (Fisher Scientific, Loughborough, UK).

### 2.7. Electrochemical Measurements

Electrochemical measurements were performed using a conventional three-electrode system, with an Ag/AgCl reference electrode and coiled platinum wire as the counter electrode, attached to a potentiostat (VersaSTAT4, AMETEK, Leicester, UK). The working electrode was a glassy carbon rotating disc electrode (RDE, Pine research) with a diameter of 5 mm (0.196 cm^2^), coated with the catalyst. All potentials are quoted relative to the RHE using the following relationship: *E*_RHE_  =  *E*_Ag/AgCl_  +  0.197  +  0.059 pH. The RDE was polished with a microfiber polishing cloth and 0.05 μm alumina slurry to obtain a mirror finish, prior to all measurements. In total, 10 mg of the catalyst was mixed with deionized water, isopropanol (99%, Aldrich, Merck, Darmstadt, Germany), and Nafion (5%, Aldrich) with a volume ratio of 9:10:1, respectively, to give an ink with a final catalyst concentration of 5 mg mL^−1^. After ultrasonication in an ice bath for 1 h, 10 μL of the well-mixed ink was drop-cast onto the RDE and dried under ambient conditions, resulting in a final loading of 50 μg of catalyst powder (~0.255 mg cm^−2^). Ohmic losses within the system were compensated for by applying an IR correction. The uncompensated system resistance was obtained using electrochemical impedance spectroscopy (EIS) at the open-circuit potential. EIS was measured over a frequency range of 1 Hz–100 kHz, with a perturbation of 10 mV. The system resistance was taken from the *x*-intercept of the Nyquist plot. The Nyquist plot during the hydrogen evolution reaction (HER) was based on EIS measurements performed in 0.5 M H_2_SO_4_, at pH = 9 with an overpotential of 30 mV over a frequency range of 10^–2^–10^6^ Hz with 10 mV sinusoidal perturbations. Linear sweep voltammetry was conducted in N_2_-saturated 0.5 M H_2_SO_4_, pH = 9, using a scan rate of 10 mV s^−1^ and electrode rotation speed of 1600 rpm. To obtain the current density, the current was normalized to the geometric electrode area, of 0.196 cm^2^. Chronopotentiometry measurements were conducted at 10 mA cm^−2^ with an RDE rotation speed of 2000 rpm to minimize the accumulation of bubbles, in order to evaluate the long-term HER stability for 20,000 s (~6 h).

## 3. Results and Discussion

### 3.1. Characterization of Laser-Reduced ZIF8@ZIF67

Samples of the ZIF-8@ZIF67 composite, LIA-ZIF8@ZIF67, and Pt-LIA-ZIF8@ZIF67 were synthesized as described above and then analyzed using STEM, Raman, and XPS. The quality of the supporting LIA-ZIF8@ZIF67 plays an important role in the catalytic performance by building up an electrical conduction network. Figure 2 shows SEM and STEM images of before and after the treatment using the IR and UV lasers.

The ZIF8@ZIF67 composite prior to the IR laser treatment shows the typical structure of ZIF-based ZIF8@ZIF67 [10]. The SEM images of the ZIF8@ZIF67 after laser treatment, however, show a significant expansion and change in structure of the ZIF8@ZIF67 composite. The IR laser treatment results in a nanoporous, rather amorphous structure. There is also evidence of spherical nanosized features on the surface of the material as shown in Figure 2b. Following the subsequent UV laser treatment, there is further loss of structure with an increase in the number of the nanosized spherical structures, highlighted by the red ellipse in Figure 2c. Figure 2d also shows the presence of roughly spherical dark features, of diameter 5–10 nm, surrounded by some periodic structure where carbon–carbon planes are separated by ~0.34 nm. The carbon interplanar spacing in pristine single-crystal graphite is 0.335 nm; a reduction in the number of layers and increase in interplanar C spacing to 0.352 nm suggest a transition to graphene [23]. We note, however, that bending of the material can also lead to an increase in the interplanar spacing, and it is clear in the STEM images in Figure 2, and Appendix A, that there is substantial bending of the C-based material around the ZIF8@ZIF67-derived metal nanoparticles. These layered structures appear to wrap around the nanosized particles. The interplanar spacing observed here is very similar to that of graphite, suggesting that the laser treatment forms graphitic carbon, rather than graphene or amorphous C. (The darker features are associated with denser material; so, they are assigned to metallic particles of Co or Zn, encapsulated by a carbonaceous shell following the IR laser treatment. This then suggests that the laser treatment leads to the formation of core–shell structures in agreement with earlier work [17,18,19]).

Figure 3 shows the elemental composition of representative LIA-ZIF8@ZIF67 samples determined by scanning transmission electron microscopy and energy dispersive X-ray spectroscopy (STEM-EDS), indicating successful loading of the LIA-ZIF8@ZIF67 with cobalt and indicating that it remains after the first LIA treatment. It is also interesting to note that the Co seems to cluster, forming particles of ~ 10–30 nm, whereas the Zn remains much more well dispersed, with particle sizes of much less than 3 nm according to the EDS mapping.

Raman spectra in Figure 4a give an indication of the structure and chemistry of the major carbonaceous structures of the LIA-ZIF8@ZIF67. The Raman spectra of the four samples show two major bands located at 1340 cm^−1^ and 1480 cm^−1^, which correspond to the defect structure (D band) and stretching of the C-C bond in graphitic sp^2^ carbon materials (G band). The intensity ratios of D to G bands (I_D_/I_G_) decreased from 0.71 to 0.61 with increased laser power; the reason is likely to be the loss of functional groups and non-carbon elements [20]. As the laser power increases, the carbonization temperature also increases, so that more functional groups and non-carbon elements present in the ZIF8@ZIF67 are eliminated, leading to an increase in the degree of graphitization and a more crystalline and ordered carbon structure. The XRD pattern of the LIA-ZIF8@ZIF67 in Figure 4b shows three characteristic peaks of metallic Co (PDF 00-015-0806) at 44.1°, 51.7°, and 75.8°, respectively; also, the carbon peak of 26.3° (PDF 00-041-1487) indicates that the ZIF8@ZIF67 is successfully carbonized.

X-ray photoelectron spectroscopy (XPS) was carried out to analyze the elemental composition and chemical states of the LIA-ZIF8@ZIF67. Survey XPS spectra are shown in Appendix A. Figure 5a shows the successful carbonization of ZIF-8@ZIF-67 after the LIA treatment. The binding energies of the C 1s peaks are shown in Table 1. The spectrum recorded from the pure ZIF8@ZIF67 (line (iii)) in Figure 5a is fitted with roughly symmetrical peaks consistent with the imidazolium framework and the presence of some adventitious carbon [21]. Following treatment with the IR laser (line (ii)) in Figure 5a, the spectrum becomes dominated with a strong asymmetric peak at a binding energy of 284.3 eV, consistent with the presence of graphitic C [24]. Some imidazolium-derived peaks remain at higher binding energies, and this is also reflected in the N 1s spectra shown in Figure 5b where the two peaks at binding energies of 399.7 eV and 400.3 eV arising from and unbound surface pyridinic N [25] and N-Co/Z-Zn [26] from the imidazolium-based framework become much weaker following the laser treatment. A third feature at a binding energy of 401.7 eV also appears. This is consistent with the formation of -NO_2_ and nitrite at the surface [27]. Treatment with the 5 mmol L^−1^ H_2_PtCl_6,_ followed by irradiation with the UV marker laser, did not result in any further significant changes to the C 1s spectrum (line (i) Figure 5a, or the N 1s spectrum (line (i) Figure 5b)). Interestingly, the C 1s spectrum recorded from the commercial Pt-C catalyst (line (iv) in Figure 5a) shows a very similar spectrum to the laser-treated samples, in that it is dominated by the sp^2^ carbon peak. The main difference is the larger sp^3^-C-derived peak, which may be due to higher levels of adventitious C adsorbed on this material. Returning to the N 1s spectra in Figure 5b, it is noted that the IR laser treatment also leads to an increase in free pyridinic N relative to the Co-N/Zn-N-derived peak. This is consistent with the observation above that the metal components aggregate to form nanoparticulate structures. The presence of the pyridinic N in the LIA-ZIF8@ZIF67 is also interesting since pyridinic N is thought to help to accelerate electron transfer and increase the diffusion-limited current, notably enhancing the catalytic activity of HER materials [28].

The XPS spectra of Zn 2p are presented in Figure 5c and consist of two spin–orbit split peaks at binding energies of 1022.3 eV and 1045.3 eV, corresponding to the Zn 2p_3/2_ and 2p_1/2_, respectively. The binding energy difference between Zn(II) and metallic Zn is too small to determine the oxidation state directly from these spectra. These binding energies are consistent with the formation of oxidized Zn at the surface of the material [29]. After irradiation, the Zn signal is seen to decrease significantly. We attribute this to the formation of the core–shell structures, so that Zn that was previously in the ZIF8@ZIF67 structure becomes wrapped in the porous C and becomes buried beneath the surface of the graphitic network. Since the kinetic energy of Zn 2p photoelectrons is around 500 eV, the XPS sampling depth will be around 3 nm. This then suggests that a reasonable amount of carbon-based material sits above the encapsulated Zn.

**Figure 5 nanomaterials-15-00078-f005:**
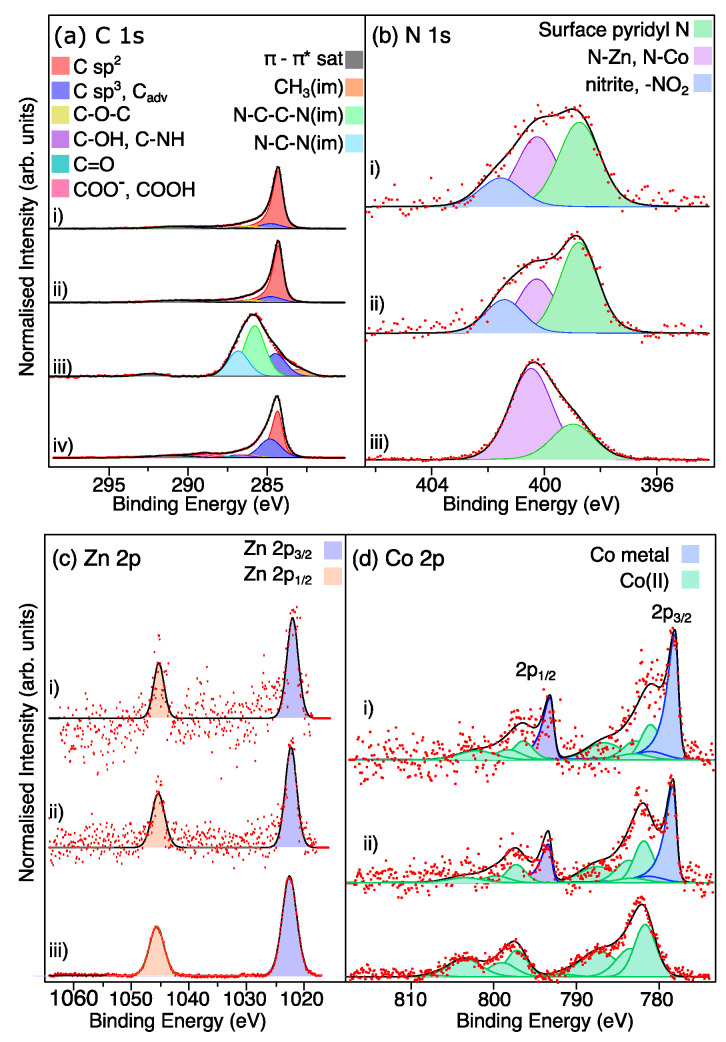
(**a**–**d**) High-resolution XPS spectra of C 1s, N 1s, and Zn 2p and Co 2p, respectively. Red dots are the raw data and black lines fit to the data from summing the components indicated by the shaded peaks. In each case, (i) is the LIA-ZIF8@ZIF67 sample, which was treated with the 1064 nm laser, Pt precursor, and UV marker laser; (ii) is the 1067 nm-irradiated LIA-ZIF8@ZIF67; (iii) is the as-prepared ZIF8@ZIF67; and (iv) is the commercial Pt-C catalyst sample. C_adv_ = adventitious carbon, im = imidazole. Co 2p spectra were fitted using the parameters and constraints by Biesinger et al. [30].

The Co 2p spectrum recorded from the ZIF8@ZIF67 in Figure 5d, before laser treatment, shows spin–orbit splitting with the main peaks centred around B.E. of 782.0 eV and 797.6 eV, arising from Co 2p_3/2_ and Co 2p_1/2_, respectively. The Co 2p_3/2_ region of this spectrum can be fitted well using the multiplet splitting associated with Co(II) species, according to the procedure of Biesinger et al. [30] in Figure 5d (line (iii)). Following irradiation with the IR laser, the first thing to notice is the decrease in the intensity of the peak, leading to higher noise levels in the data. This agrees with the intensity in the Zn 2p spectrum and again arises due to the encapsulation of the Co with graphitic carbon due to the carbonization of the ZIF8@ZIF67 and aggregation of the Co to form nanoparticles. In addition, two strong spin–orbit split features at binding energies of 778.5 eV and 793.5 eV can be observed in the Co 2p spectrum. These are consistent with the formation of metallic Co, and are fitted with asymmetric line shapes, further suggesting the metallic nature of the Co giving rise to these features.

### 3.2. Characterization of Pt Single Atoms

Following treatment with the chloroplatinic acid and treatment with the UV laser, the Pt-LIA-ZIF8@ZIF67 samples were characterized using aberration-corrected high-angle annular dark-field scanning transmission electron microscopy (STEM-HAADF). Figure 6 shows the STEM-HAADF of the Pt-LIA-ZIF8@ZIF67 sample. The STEM-HAADF images indicate that Pt nanoparticles are not formed. Instead, we see the presence of Pt single atoms, as bright spots, dispersed throughout the Pt-LIA-ZIF8@ZIF67. In Figure 7, the STEM-coupled EDS elemental mapping confirms the presence of Pt, Co, Zn, C, and N in the powder, and also shows a uniform distribution of Pt on Co, Zn, and the Pt-LIA-ZIF8@ZIF67.

The XPS Pt 4f high-resolution spectra provide further evidence for the singly atomic dispersion of Pt in the Pt-LIA-ZIF8@ZIF67 sample, as shown in Figure 8a. The spin–orbit split Pt 3d_5/2_ and 3d_3/2_ peaks for the CM Pt/C are located at binding energies of 71.4 eV and 74.7 eV, corresponding to the doublet peaks of metallic Pt 4f_7/2_ and 4f_5/2_, respectively [31]. These peaks are fitted with asymmetric line shapes to account for the metallic nature of the nanoparticulate Pt [32]. For the laser-treated Pt-LIA-ZIF8@ZIF67, however, the doublet peaks are shifted to higher binding energies of 72.6 eV (4f_7/2_) and 75.8 eV (4f_5/2_). This is due to the smaller particle size resulting in poorer screening of the core hole, which results in a shift to higher binding energy [31]. In addition, the peaks for the Pt-LIA-ZIF8@ZIF67 are broadened and symmetrical, suggesting that the Pt here is not metallic. The lack of a detectable metallic Pt signal implies that the Pt nanoparticles or clusters are unlikely to exist or have a negligible content in the sample, in agreement with the STEM-HAADF images in Figure 6.

The absence of crystalline Pt phases in the Pt-LIA-ZIF8@ZIF67 is further supported by the measured XRD patterns, which are compared to the patterns calculated using Mercury software [33,34,35], and a Pt CIF file, shown in Figure 8b. No crystalline phase of Pt is observed in the XRD pattern, since the characteristic peaks of Pt (111), (200), (220), and (311) that are present in the XRD pattern of commercial Pt/C, and the calculated powder XRD pattern, are absent in the XRD pattern recorded from the Pt-LIA-ZIF8@ZIF67 sample. We note that the XRD peaks of the commercial Pt-C are broad, in agreement with the nanoparticle structure of the commercial catalyst. XRD patterns shown in Appendix A, also show that the broadened, symmetrical peaks, and the shift to higher binding energy observed for the Pt 4f peaks in the XPS spectra, are not due to the formation of PtO_2_.

### 3.3. Electrocatalytic Performance

The electrocatalytic HER activity of the Pt-LIA-ZIF8@ZIF67 samples irradiated with the UV marker laser was evaluated using a standard three-electrode system with a rotating disc electrode (RDE). The measurement was conducted in a N_2_-saturated 0.5 M H_2_SO_4_ electrolyte. To determine the catalytic performance, an ohmic-drop correction was conducted to minimize the effect of solution resistance. Linear sweep voltammogram (LSV) curves of the commercial Pt/C catalyst (20 wt. %; Highspec3000, Johnson Matthey, Royston, UK) were also measured for comparison. Note that the Pt1-LIA-ZIF8@ZIF67 and Pt5-LIA-ZIF8@ZIF67 have Pt species in the form of single atoms, while the Pt10-LIA-ZIF8@ZIF67 has both single atoms and clusters present in the material.

Using the thermodynamic HER potential (H ^+^ /H_2_ = 0 V vs. RHE) as a reference, Figure 9a shows that the Pt10-LIA-ZIF8@ZIF67 and Pt5-LIA-ZIF8@ZIF67 samples exhibit small overpotentials of 56.7 mV and 69.8 mV, respectively, to drive a cathodic current density of 10 mA cm^−2^. These are comparable to that of the commercial Pt/C catalyst (30.5 mV). Decreasing the Pt loading to ~0.17%, the Pt1-LIA-ZIF8@ZIF67 sample shows an overpotential of 141.3 mV at 10 mA cm^−2^, indicating a reduction in the number of active sites on the LIA-ZIF8@ZIF67 support. The HER catalysis kinetics were assessed using Tafel plots, shown in Figure 9b. The measured Tafel slopes of Pt10-LIA-ZIF8@ZIF67 (49.8 mV dec^−1^) and Pt5-LIA-ZIF8@ZIF67 (108.08 mV dec^−1^) are comparable to the CM Pt/C (98.76 mV dec^−1^), despite the low Pt loading (2.18 wt. % Pt and 0.86 wt. % Pt vs. 20 wt. % Pt in the commercial sample), which suggests that the kinetics of the HER are superior to those of the commercial catalyst studied here, in the short term. As expected, the Pt1-LIA-ZIF8@ZIF67 shows a large Tafel slope of 159.8 mV dec^−1^ due to a lack of active sites.

Normalizing the measured current, at a given overpotential, to the Pt mass loading allows for the mass activities of the catalysts to be evaluated. An overpotential of 50 mV was chosen for these measurements to compare with literature-reported values. As shown in Figure 9c, because of the significantly lower Pt loading in the Pt5-LIA-ZIF8@ZIF67 (0.86 wt. % Pt) vs. the CM Pt/C (20 wt. % Pt), the Pt5-LIA-ZIF8@ZIF67 mass activity is an order of magnitude higher (18.12 A mg^−1^) compared to the CM Pt/C (0.88 A mg^−1^). This increased HER mass activity for the Pt5-LIA-ZIF8@ZIF67, also outperforms the majority of literature values [2,6,12,36,37,38,39,40]. This suggests high Pt utilization and great promise for practical applications, from these short-term measurements. Compared to the Pt5-LIA-ZIF8@ZIF67, the Pt10-LIA-ZIF8@ZIF67 shows a moderate enhancement in mass activity, which is likely to be due to a fraction of inert Pt atoms becoming embedded inside the clusters.

The influence of the metal precursor concentration on the HER performance of the Pt-LIA-ZIF8@ZIF67 was also investigated. The overpotentials at 10 mAcm^−2^ of Pt1-LIA-ZIF8@ZIF67, Pt5-LIA-ZIF8@ZIF67, and Pt10-LIA-ZIF8@ZIF67 were 141.3 mV, 69.8 mV, and 56.7 mV, respectively, demonstrating that the HER activity and kinetics vary as a function of metal precursor concentrations despite using the same laser outputs under these fabrication conditions.

In order to examine the long-term performance of the catalysts, the potential vs. RHE for Pt5-LIA-ZIF8@ZIF67 was measured for ~6 h at a fixed current density of 10 mA cm^−2^, and an RDE rotation speed of 2000 rpm. The electrolyte was purged with N_2_ prior to the measurement. As shown in Figure 10, although the potential relative to RHE is similar for the SAC and the commercial catalyst at *t* = 0, there is a rapid drop in the potential of the SAC over the first ~4500 s to −0.390 V. The potential then stabilizes somewhat and drops to −0.395 V after 20,000 s. The commercial Pt/C catalyst on the other hand exhibits only a slight linear decrease in the potential. Since the Pt loading of the SAC is only 0.87%, and the commercial Pt/C catalyst is 20 wt. %, then even after 20,000 s, the mass activity of the Pt5-LIA-ZIF8@ZIF67SAC synthesized here is still higher than that of the commercial catalyst.

The reasons for the rapid decrease in the activity is not clear, and it is difficult to characterize the used catalyst since only 5 mg of the catalyst is used, and it is mixed with isopropanol and Nafion. This makes it extremely difficult to obtain TEM images from the used catalyst. One possibility for the fall in activity is that the Co and Zn nanoparticles react with the H_2_SO_4_ electrolyte, which changes the surface chemistry of the nanoparticles. STEM HAADF and EDS images of the Pt5-LIA-ZIF8@ZIF67 sample, following washing of the catalyst in 0.5 M H_2_SO_4_, are shown in Figure 10b–d. Appendix A shows a second Pt5-LIA-ZIF8@ZIF67 sample that underwent the same acid wash treatment. EDS of Co is not shown in Figure 10 as none were found in this sample, suggesting that the acid removed the Co, although in Appendix A, there is still some Co signal in the EDS scans. The images may indicate that a change in the non-noble metal composition, due to a reaction with the acid, may be responsible for the loss in activity. It is also clear from Figure 10d and Appendix A that there is some clustering of the Pt to form nanoparticles, which would reduce the number of active sites. Figure 10 shows that the increased Pt aggregation appears to be linked to a higher Zn concentration, but it is not clear whether the coalescence of the two elements here is correlated. In Appendix A, on the other hand, although there is less obvious aggregation of the Pt, there do seem to be some slightly larger particles. Although these two phenomena are likely to play a part in the rapid fall off in the catalytic activity, the graphitic support may also play a part. It is known, for example, that the presence of H_2_ bubbles produced in the HER process can cause damage to the basal plane of graphene [41]. The conductive pathways of 3D-interconnected graphene substrates can be destroyed after long-term testing, thus leading to the degradation of electrocatalytic activity in HER. Therefore, graphene substrates with fewer structural defects and better mechanical properties would be expected to exhibit superior stability during HER testing. According to the Raman results shown in Figure 4, the LIA-ZIF8@ZIF67 synthesized here shows fewer structural defects at higher laser fluences, but unfortunately the EDS maps do not offer insight into the effects of the acid washing on the graphite structure. Therefore, future work will investigate the longer-term activity of LIA-ZIF8@ZIF67 synthesized at higher laser fluences, to determine whether they are able to bear the hydrogen evolution perturbation more effectively, and whether this treatment may also lead to less Pt clustering upon exposure to the acidic electrolyte.

## 4. Conclusions

In summary, we proposed a two-step laser-induced annealing ZIF8@ZIF67 to uniformly disperse the Pt single atoms with ultrahigh mass activity efficiency for HER. The pronounced photothermal effect of the 1064 nm laser irradiation in combination with fast scanning enables a high processing temperature with a short heating time and rapid quenching time, guaranteeing the full reduction of ZIF8@ZIF67; but, also, the 355 nm UV laser system prevents the migration of metal atoms of Pt ions. We demonstrated that the Pt-LIA-ZIF8@ZIF67 exhibits promising activity and high Pt utilization in HER with an overpotential of 69.8 mV at 10 mA cm^−2^ and mass activity of 18.12 A/mg Pt at 50 mV. The use of a laser-based technique is an economical and straightforward method that offers a novel route to produce precious metal electrocatalysts with exceptional short-term efficiency for energy conversion and storage purposes. Although the long-term activity appears to decline rather rapidly, further work aims to determine the key mechanisms behind the degradation of the catalytic activity, and therefore methods to mitigate this.

## Figures and Tables

**Figure 1 nanomaterials-15-00078-f001:**
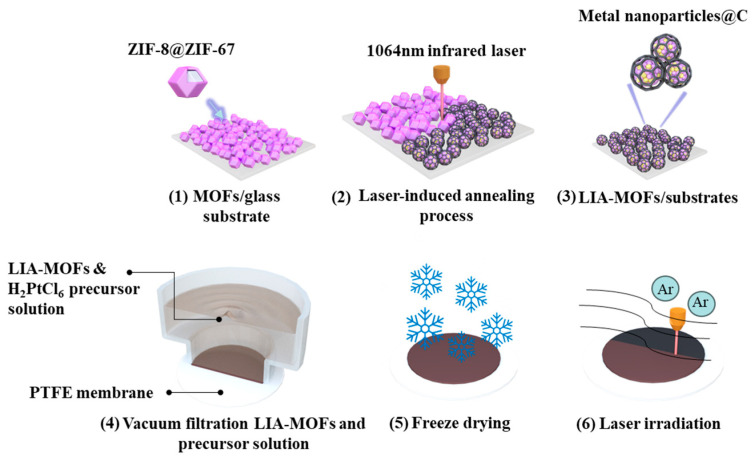
(**1**) ZIF8@ZIF67 deposited on glass substrate; (**2**) ZIF8@ZIF67 under 1064 nm infrared laser-induced annealing process; (**3**) LIA-ZIF8@ZIF67 coated onto glass substrates; (**4**) preparation of LIA-ZIF8@ZIF67 with Pt metal precursor hydrogel film; (**5**) freeze-drying process; (**6**) direct UV laser irradiation to form Pt-LIA-ZIF8@ZIF67.

**Figure 2 nanomaterials-15-00078-f002:**
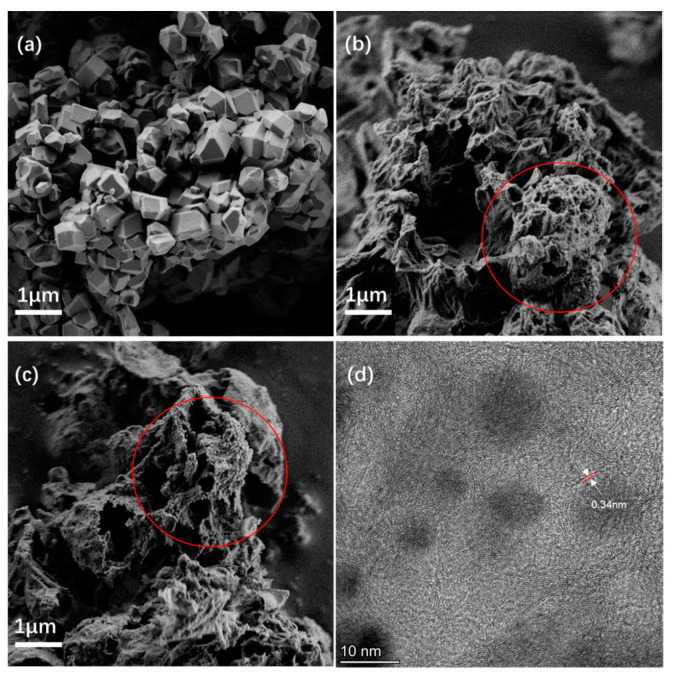
(**a**) An SEM image of ZIF-8@ZIF-67 with no laser treatment; (**b**) SEM after IR laser treatment with a 1067 nm laser (LIA-ZIF8@ZIF67); (**c**) SEM after IR and UV (355 nm) laser treatment with Pt single atoms. The red circles in (**b**,**c**) highlight some nanosized particles on the surface of the laser-irradiated ZIF8@ZIF67; (**d**) an STEM image of the ZIF-8@ZIF-67 after the first IR laser treatment showing the formation of a core–shell structure: the dark regions are due to ZIF8@ZIF67-derived metal nanoparticles formed by the laser treatment, and fringes surrounding them arose from the C support. A zoomed in region of (**d**) is shown in Appendix A.

**Figure 3 nanomaterials-15-00078-f003:**
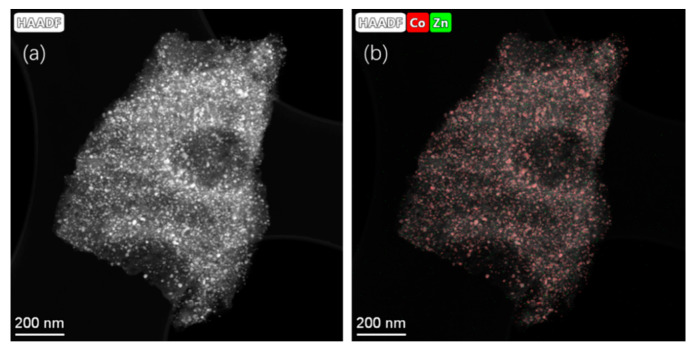
(**a**) STEM-HAADF images and (**b**) EDS mapping using Co K-edge and Zn K-edge after poly-nominal background subtraction of ZIF-8@ZIF-67 after IR laser treatment. Associated spectrum is shown in Appendix A.

**Figure 4 nanomaterials-15-00078-f004:**
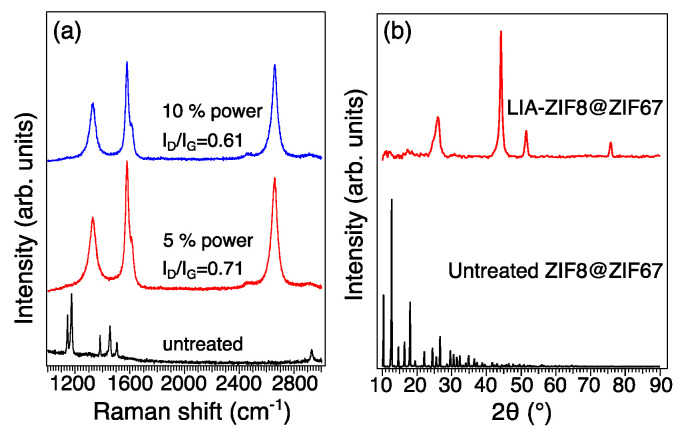
(**a**) Raman spectroscopy and (**b**) XRD patterns of untreated ZIF8@ZIF67 and LIA-ZIF8@ZIF67.

**Figure 6 nanomaterials-15-00078-f006:**
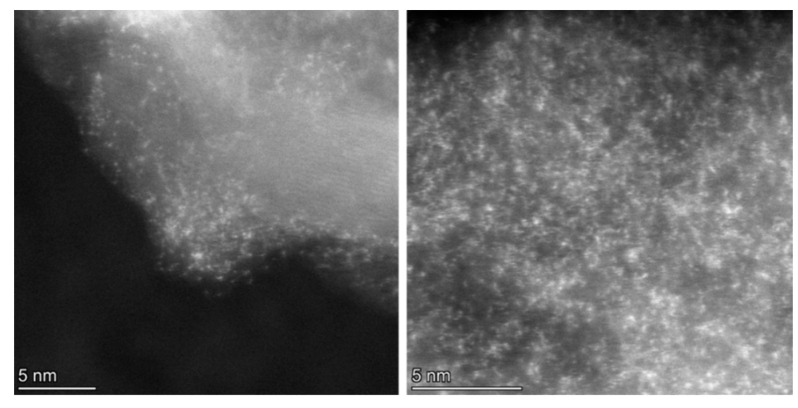
STEM-HAADF images of Pt single atoms, which appear as bright spots with an average diameter of ~2 Å.

**Figure 7 nanomaterials-15-00078-f007:**
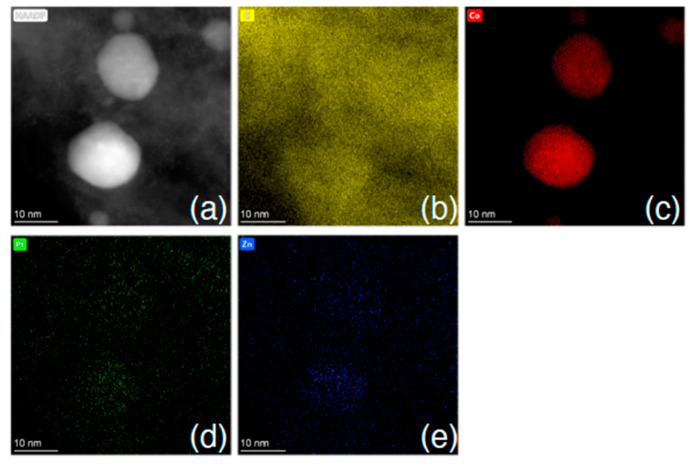
(**a**) STEM-HAADF image and STEM-EDS element mapping of (**b**) C K-edge, (**c**) Co K-edge, (**d**) Pt L-edge, and (**e**) Zn K-edge after poly-nominal background subtraction. Associated spectrum is shown in Appendix A.

**Figure 8 nanomaterials-15-00078-f008:**
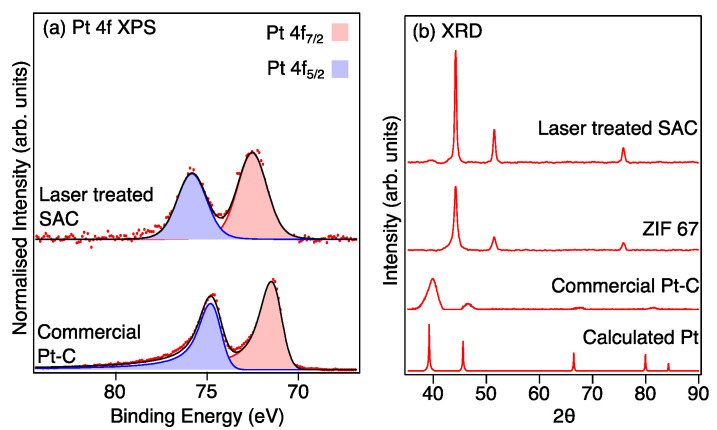
(**a**) Pt 4f XPS recorded from the commercial Pt-C catalyst and the sample prepared from the ZIF8@ZIF67 after irradiation with a *l* = 1067 nm laser, treatment with a Pt precursor (H_2_PtCl_2_), and further irradiation with a UV marker laser (*l_centre_* = 355 nm). Red dots are the raw data and the black line is a fit to the data. (**b**) XRD patterns of the bulk platinum (calculated from Mercury software [30]), commercial Pt-C catalyst, and ZIF8@ZIF67 after irradiation with the 1067 nm laser (LIA ZIF8@ZIF67), and the LIA ZIF8@ZIF67 after further treatment with a Pt precursor and the UV marker laser. The Pt-LIA shows no sign of crystalline Pt in the XRD pattern, despite a clear signal in the XPS spectrum.

**Figure 9 nanomaterials-15-00078-f009:**
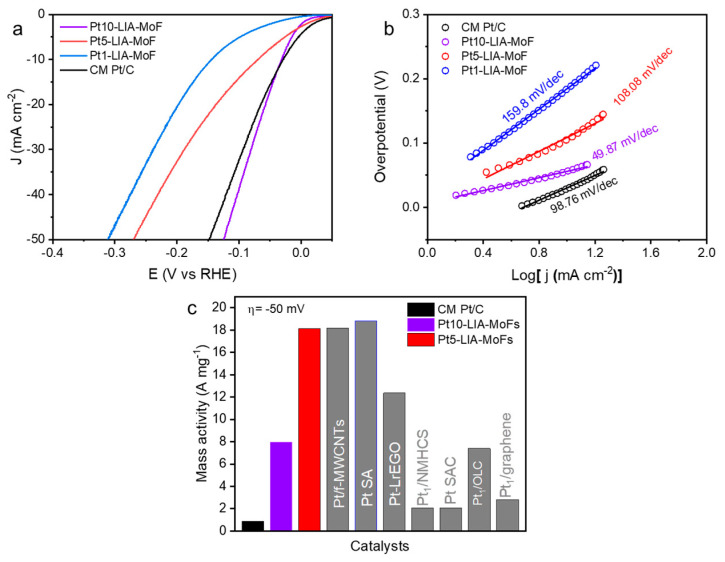
(**a**) LSVs (10 mV s^−1^; after ohmic-drop correction) of the commercial Pt/C (20 wt. %) and the Pt-LIA-ZIF8@ZIF67 catalysts prepared by the laser irradiation of the LIA-ZIF8@ZIF67 infused with Pt precursor solutions at different concentrations; the LSVs were recorded in N_2_-saturated 0.5 M H_2_SO_4_ at a rotation speed of 1600 rpm. (**b**) Tafel slopes of the commercial Pt/C and Pt10-LIA-ZIF8@ZIF67, Pt5-LIA-ZIF8@ZIF67, and Pt1-LIA-ZIF8@ZIF67. (**c**) Mass activities of the commercial Pt/C, Pt5-LIA-ZIF8@ZIF67, Pt10-LIA-ZIF8@ZIF67, and the literature-reported Pt-based SACs at an overpotential of 50 mV.

**Figure 10 nanomaterials-15-00078-f010:**
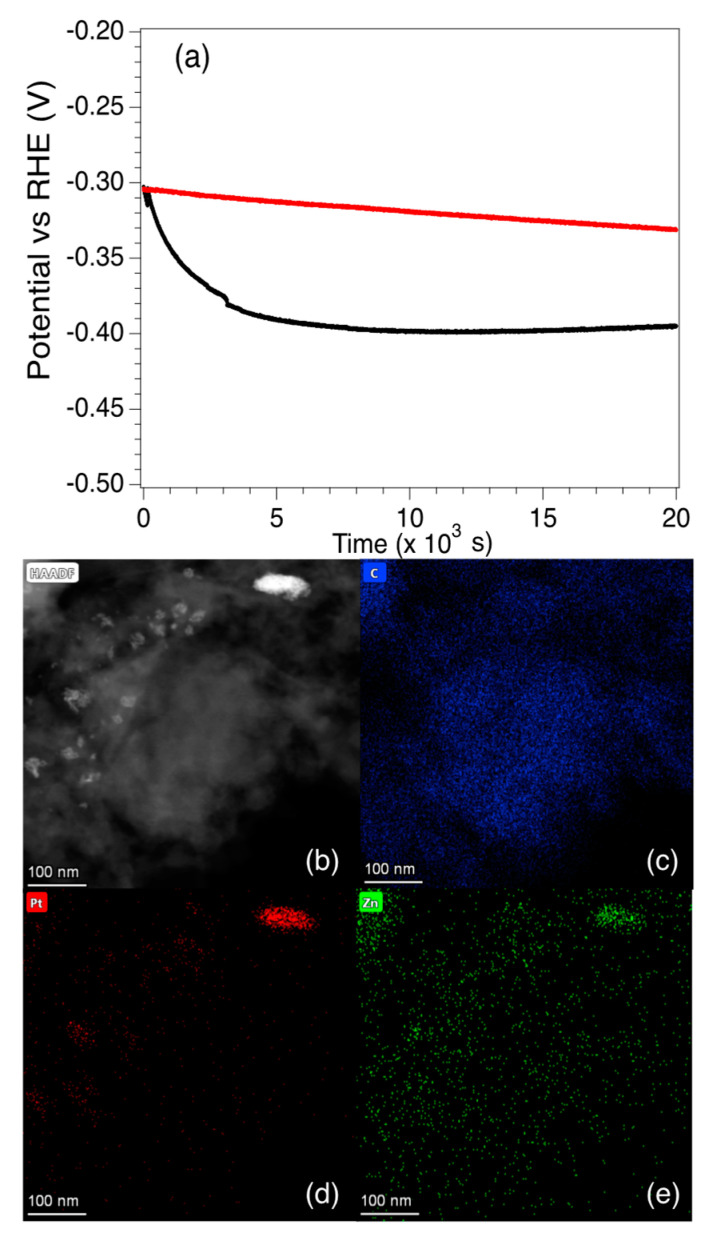
(**a**) Chronopotentiometric behaviour of the commercial Pt/C catalyst and the Pt5-LIA-ZIF8@ZIF67 over ~6 h. At t = 0, both catalysts have a potential vs. RHE of −0.302 V. The potential of the commercial catalyst falls to −0.331 V after 6 h. The SAC Pt5-LIA-ZIF8@ZIF67 shows a rapid drop in potential to ~−0.390 V at t = 4460 s, after which the decay is much slower, and reaches a value of −0.395 V at 20 × 10^3^ s. (**b**) An HAADF image of Pt5-LIA-ZIF8@ZIF67. EDS scans over (**c**) C, (**d**) Pt, and (**e**) Zn are also shown. It is clear that in this specimen, the Pt begins to cluster around a feature at the top right of the image. Zn accumulation is also observed in this region. No Co was found in this image, nor in the EDS spectrum recorded from this sample as shown in Appendix A.

**Table 1 nanomaterials-15-00078-t001:** Binding energies (B.E.) and peak assignments of features in the C 1s spectra of the commercial catalyst, ZIF8@ZIF67, and laser-annealed and Pt-treated ZIF8@ZIF67. All binding energies are quoted as ±0.2 eV.

Commercial Pt-C		ZIF67@ZIF8		IR Laser-Treated		IR + UV + Pt-Treated	
B.E. (eV)	Species	B.E. (eV)	Species	B.E. (eV)	Species	B.E. (eV)	Species
284.4	sp^2^ C	283.3	sp^2^ †	284.4	sp^2^ C	284.3	sp^2^ C
284.8	sp^3^ C	284.8	sp^3^ C	284.8	sp^3^ C	284.7	sp^3^ C
286.1	C-O-C	286.2	NCCN †	286.2	C-O-C	286.1	C-O-C
286.7	COH/CNH	287.2	N-C-N †	286.7	COH/CNH	286.6	COH/CNH
287.9	C=O	292.8	p-p* sat	287.9	C=O	288.0	C=O
289.1	COOR ^#^			289.1	COOR	289.1	COOR
291.1	p-p* sat			290.9	p-p* sat	291.0	p-p* sat

† arising from the imidazole framework. ^#^ R = H, C.

## Data Availability

Raw data and images are available free of charge from http://doi.org/10.3390/nano15010078 or directly from the authors.

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
