# Peer review of "Laser Synthesis of Platinum Single-Atom Catalysts for Hydrogen Evolution Reaction"

_nanomaterials, 2025, doi:10.3390/nano15010078_

Round 1

Reviewer 1 Report

Comments and Suggestions for Authors

In this manuscript, the authors reported a novel two-step annealing strategy for the synthesis of Pt single-atom catalysts on carbon substrate and their application toward the hydrogen evolution reaction (HER). Overall, this work has high novelty and the data were clearly presented. I had very much enjoyed reading this work. The manuscript should be suitable for the journal Nanomaterial. However, some minor comments, as detailed below, should be properly addressed to further enhance the quality and clarity of this work.

1. The electrochemical HER activity was not consistently provided, in line 24 it was “−68.8 mV at −10 mA cm-2” while in line 432 it was “69.8 mV at 10 mA/cm2”. Please ensure consistency.

2. Recent works on HER, SACs and MOFs are suggested to be referenced in the Introduction (e.g., DOI: 10.1016/j.matre.2022.100144; doi: 10.1016/j.matre.2023.100215)

3. Line 256, “Survey XPS spectra are shown in Supporting Information Figure SX.” However, no survey XPS spectra were found in the manuscript.

4. Line 223, “the planes are separated by ~0.34 nm”, what do this distance refer to? Please provide more discussion if applicable.

5. Line 201, “Chronopotentiometry measurements (at 10 mA cm-2) were conducted to evaluate the long-term HER stability”. However, no such data were found in the manuscript.

6. Figure S4 and S5 were not mentioned in the main text. How were they related to this work?

7. There is a comment in the Supplementary Information.

Reviewer 2 Report

Comments and Suggestions for Authors

This work reports a two-step laser annealing strategy for synthesizing platinum single-atom catalysts (Pt SACs) on carbon substrates. The process involves fast laser scanning/irradiation of freeze-dried metal-organic framework (MOF) films (ZIF67@ZIF8 composites) using an ns pulsed infrared (IR; 1064 nm) laser to produce a metal-loaded graphitized film. Subsequently, chloroplatinic acid (H₂PtCl₆) is added to this film, which is then irradiated with ultraviolet (UV; 355 nm) laser light. This step pyrolyzes H₂PtCl₆ to form single-atom catalysts (SACs) and further reduces/graphitizes the MOF to create Pt-LIA-MOF. The resulting catalyst was found to be 20.52 times more mass-active than commercial Pt/C catalysts at a 50 mV overpotential versus the reversible hydrogen electrode (RHE). Despite some intriguing results, after careful review, I cannot recommend the publication of this work in its current form. Several concerns need to be addressed:

1. Figure 2d should be magnified. The lattice stripes are not very visible in this figure, and a zoomed-in view would make it more intuitive.

2. The roughness of the graphs makes them difficult to read. For example, there is an inconsistency in the font of the horizontal and vertical coordinates in Figure 4a and 4b. Additionally, Figure 4b should read 'MOFs' instead of 'MoFs'.

3. In Figure 5d, the raw data for Co (red dots) clearly shows a distinct peak between 810 eV and 790 eV, but the figure has not been fitted. Please revise the data processing and label the attribution of the peak.

4. Some textual details need to be standardized. For instance, the notation 'Zn 2p₃/₂ and 2p₁/₂' in line 282 should be consistent. The red markings 'Error! Reference source not found.' in lines 208 and 335 should be corrected, and relevant literature should be cited properly.

5. Citations should be in a consistent format. For example, in reference [24], the subscripts in 'TiO₂' and 'SO₂' should be formatted correctly. The year in reference [32] should be in bold.

6. If the focus is on demonstrating the superiority of this synthesis method, a comparison with multiple synthesis methods should be conducted.

Comments on the Quality of English Language

English and formats should be polished.

Reviewer 3 Report

Comments and Suggestions for Authors

In this study, the authors synthesized Pt single-atom-dispersed MOF-derived carbon-based materials as acidic hydrogen evolution electrocatalysts, which is interesting. However, I have some concerns that the authors need to address.

1. The authors need to conduct long-term HER testing of their electrocatalysts.

2. The authors need to check the pH value of their electrolyte and include it in the main manuscript.

3. Throughout the main manuscript, I found error descriptions. For example, the authors need to check "Lines 134-135 on Page 3," "Line 208 on Page 5," "Line 256 on Page 8," "Line 335 on Page 11," etc. The authors need to double-check the main manuscript to minimize errors.

Comments on the Quality of English Language

Several Figure captions are unclear (Figure S4, Figure S5, Figure S6, etc.). The authors should thoroughly review and revise all captions for accuracy and clarity.

Round 2

Reviewer 3 Report

Comments and Suggestions for Authors

The authors addressed almost all my concerns in the first review round. However, in the long-term HER testing, their catalyst experienced drastic activity decay within ~ two hours. The authors need to reveal the reason for this phenomenon by conducting characterizations (e.g., STEM-EDS element mapping, XPS, etc.) of the post-HER catalyst.

Additionally, this manuscript still contains minor errors. The authors need to double-check the manuscript again.

Author Response

The authors addressed almost all my concerns in the first review round. However, in the long-term HER testing, their catalyst experienced drastic activity decay within ~ two hours. The authors need to reveal the reason for this phenomenon by conducting characterizations (e.g., STEM-EDS element mapping, XPS, etc.) of the post-HER catalyst.

We thank the reviewer for the comment here, and acknowledge that this would indeed be useful. The reason such characterisation was not carried out was that the catalyst used in testing was heavily contaminated by Nafion and ethanol and because of this the signal from both EDS and XPS was very weak. In fact the XPS from the used catalyst was almost entirely drowned out by the C, Na and F signals. As a 'surrogate' we have instead added EDS maps of two samples washed with 0.5 M H2SO4 for the same amount of time. We have included one set of EDS images in the main manuscript (where we see complete loss of Co and aggregation of Pt) and one set in the supplementary information, where we see some loss of Co and some less obvious clustering. These still do not clearly allow us to identify loss of the transition metals or aggregation of the Pt as a sole cause of the rapid loss in activity - it looks like both could play a part. We are however looking at the effect of laser power on the clustering of the Pt single atoms in continuing work, as well as whether there is a link between Zn clusters and Pt clusters in these materials.

Additionally, this manuscript still contains minor errors. The authors need to double-check the manuscript again.

We have checked the manuscript again and hopefully have identified all of the minor errors noted by the reviewer.

Round 3

Reviewer 3 Report

Comments and Suggestions for Authors

The authors appear to have conducted thorough work, and the manuscript is suitable for acceptance in Nanomaterials. However, prior to final acceptance, the authors are requested to include the EDX spectrum corresponding to Figures 10b-e in the supporting information.

Author Response

The authors appear to have conducted thorough work, and the manuscript is suitable for acceptance in Nanomaterials. However, prior to final acceptance, the authors are requested to include the EDX spectrum corresponding to Figures 10b-e in the supporting information.

The EDX spectrum has been added to the supporting information as Figure S6 and the figure caption to Figure 10 in the main manuscript has been updated to include this.